# Attenuated *Salmonella enterica* Serovar Typhimurium, Strain NC983, Is Immunogenic, and Protective against Virulent Typhimurium Challenges in Mice

**DOI:** 10.3390/vaccines8040646

**Published:** 2020-11-03

**Authors:** Bryan Troxell, Mary Mendoza, Rizwana Ali, Matthew Koci, Hosni Hassan

**Affiliations:** 1Prestage Department of Poultry Science, North Carolina State University, Raleigh, NC 27695, USA; sbtroxel@ncsu.edu (B.T.); mamendoz@ncsu.edu (M.M.); riz@ncsu.edu (R.A.); mdkoci@ncsu.edu (M.K.); 2Microbiology Graduate Program, North Carolina State University, Raleigh, NC 27695, USA

**Keywords:** *Salmonella* typhimurium, NC983, vaccine, in vivo, infection, C57BL/6 mice, BALB/c mice, IgG

## Abstract

Non-typhoidal *Salmonella* (NTS) serovars are significant health burden worldwide. Although much effort has been devoted to developing typhoid-based vaccines for humans, currently there is no NTS vaccine available. Presented here is the efficacy of a live attenuated serovar Typhimurium strain (NC983). Oral delivery of strain NC983 was capable of fully protecting C57BL/6 and BALB/c mice against challenge with virulent Typhimurium. Strain NC983 was found to elicit an anti-Typhimurium IgG response following administration of vaccine and boosting doses. Furthermore, in competition experiments with virulent *S.* Typhimurium (ATCC 14028), NC983 was highly defective in colonization of the murine liver and spleen. Collectively, these results indicate that strain NC983 is a potential live attenuated vaccine strain that warrants further development.

## 1. Introduction

Foodborne pathogens are a group of infectious agents that threaten public health. Members of this diverse group contaminate food and water while being spread between the environment, agriculture animals, plants, and people. The infectious agents within this group that are capable of surviving within diverse animal hosts are problematic because of their ability to induce a carrier state in certain animal hosts. An example of this is the non-Typhoid *Salmonella enterica* (NTS) serovars. *Salmonella enterica* consists of over 2500 serovars some of which exhibit strict host specificity, such as serovar Typhi (*S. Typhi*), and others exhibit broad host specificity, such as serovar Typhimurium (*S.* Typhimurium). Collectively, *S. enterica* has been linked to the majority of foodborne outbreaks within the USA [1]. The CDC estimates about 1.35 Million *Salmonella* infections, 26,500 hospitalizations, and 420 deaths per year in the USA alone. The symptoms of NTS infections range from self-limiting gastroenteritis to sepsis and death, unless treated with antibiotics [2,3]. Currently, about 8% of *Salmonella* infections are resistant to one or more class of antimicrobials [4,5]. Unfortunately, the increased incidences of antibiotic resistance among NTS infections is a serious challenge to public health.

Because of the threat to public health numerous vaccines protecting against *S. enterica* serovars have been developed and tested. Early work determined that a live attenuated strain generated superior protection against wild-type challenge compared to heat inactivation of a virulent strain [6]. Subsequently, *S. enterica* strains bearing mutations in amino acid and nucleic acid biosynthetic pathways resulting in auxotrophy for these compounds, as well as, mutations in the UDP-glucose 4-epimerase (*galE*) functioned as live-attenuated vaccines for multiple serovars [6,7,8,9,10,11,12,13]. Moreover, live attenuated vaccine strains are promising genetic backgrounds for the delivery of heterologous antigens in the vaccination against other bacterial pathogens [14,15,16,17,18,19,20,21,22,23,24,25,26,27].

Alternative approaches to using live attenuated vaccines have demonstrated immunogenicity. Recently, the immunogenicity of outer membrane vesicles and glycoconjugates vaccines were tested using the two major serovars responsible for NTS infections, Typhimurium and Enteritidis [28]. Results indicate mice generated an IgG1 antibody response to treatment. These bacterial-derived products can limit the potential risks associated with using live vaccines; however, the cultivation conditions and the genetic modulation of bacteria can impact the antigens generated in vitro. These antigens may or may not be expressed in vivo. For instance, *Salmonella* pathogenicity island 2 (SPI-2) is weakly expressed under standard in vitro conditions, but highly expressed in the *Salmonella* containing vacuole [29]. In this regard, live attenuated vaccines may provide greater efficacy since viability is maintained until the host condition and immune response restricts growth. This scenario allows for the expression of infection-relevant antigens. In addition, strains can be genetically modified to exhibit delayed attenuation allowing for an increased immune response by the host [26].

In addition to their protective role against wild-type challenge and promise as antigen delivery vehicles, live attenuated *S. enterica* vaccine strains may be useful in treating other aspects of human health (i.e., in a murine model of non-Hodgkin lymphoma [30,31], for targeting tumors in vivo [32,33,34,35,36,37,38]). Thus, the utility of live attenuated *S. enterica* vaccine strains in protection against individual serovars, delivery of heterologous antigens for protection against other infectious diseases, and as a supplement to current cancer therapies is appreciable.

Preventative measures such as vaccination with live attenuated strains may be effective at reducing the burden of NTS. Since the 1990s, there have been 45 outbreaks of salmonellosis linked to poultry [39]. 2010 saw one of the biggest and most high profile poultry related outbreaks involving 11 states, ≈1939 illnesses, and the recall of 380 million eggs [1]. In 2011, the value of all chickens in the United States was 1.7 billion dollars [40] and the economic cost of NTS was ≈2.7 billion dollars [41]. Collectively, NTS has a significant economic impact on producers, as well as, a significant economic and medical impact on consumers. However, as discussed previously, an effective vaccine will need to exhibit some characteristics of the wild-type, but not to the extent to advance disease [42]. Thus, a live attenuated vaccine strain is advantageous if tolerated, immunogenic, and protective.

During an attempt to construct a Δ*fnr* mutant in the virulent *S.* Typhimurium background (American Type Culture Collection strain, ATCC 14028s), we transduced *fnr*::Tn*10* from the non-virulent *S.* Typhimurium strain LT2 (SL2986/TN2958) to the virulent strain ATCC 14028s by using P22 phage. The transductants (14028s, *fnr::*Tn*10*) were plated on Evans Blue-Uranine agar, and the verified mutant strain was treated with fusaric acid, according to the method developed by Bruce Ames and his colleagues [43], to select for loss of the tetracycline resistant phenotype. One of the selected isolates, NC983, was shown to be attenuated in mice [44]; however, its utility as a live attenuated vaccine strain was not tested. Here, we present evidence that strain NC983 is a vaccine candidate.

## 2. Materials and Methods

### 2.1. Bacterial Strains

Table 1 lists the bacterial strains used in this study. The parental strain used in this study is from ATCC. Construction of spontaneous rifampicin resistant Typhimurium strains were generated as described previously [45]. Strain NC1040 is a kanamycin resistant derivative of 14028s that is fully virulent in mice and was constructed as described previously [45].

### 2.2. Bacterial Growth and Preparation of Cell Suspensions

NC983 or the challenge strains were grown overnight at 37 °C in ~100 mL of Luria–Bertani (LB; 10 g tryptone, 5 g yeast extract, and 10 g NaCl per L) under static culture conditions. Bacteria were centrifuged, washed in phosphate buffered saline (PBS), and resuspended in ~20 mL of PBS. The optical density at 600 nm (OD_600_) of the concentrated cell suspension was determined using a Bio-Rad Smart spec 3000 with a 1 cm light path, and adjusted, according to a standard predetermined relationship between OD_600_ and viable cell counts (i.e., 1 OD_600_ ~1 × 10^9^ CFU/mL), to an appropriate cell density as indicated in the results. The cell suspension was diluted and plated to confirm the actual viable CFU/mL.

*Salmonella* strains were enumerated on XLT agar media (Acumedia, Neogen Corp., Lansing, MI, USA) containing 100 mM MOPS and the appropriate antibiotics—Kanamycin (Kan), 65 µg/mL and Rifampicin (Rif), 100 µg/mL.

### 2.3. Animals

Six to eight-week-old C57BL/6 and BALB/c (*Ity ^S^*, both strains are *S.* Typhimurium sensitive) female mice from Jackson Laboratories (Bar Harbor, ME, USA) and Harlan Lab (now Envigo, Indianapolis, IN, USA), respectively, were used. Mice were housed in disposable cages (4 mice per cage) and had access to sterile water and food (PicoLab Mouse Diet 2) *ad libitum*.

### 2.4. Determination of Dose Required to Kill 50% of Mice (LD_50_)

The lethal dose required to kill 50% of animals (LD_50_) for *S.* Typhimurium ATCC 14028s was determined under our conditions. Four groups of mice (four mice per group) each received an oral dose of 3.5 × 10^1^, 3.5 × 10^2^, 3.5 × 10^3^, or 3.5 × 10^4^ CFU/mouse. Mice were monitored for 14 days and the LD_50_ was calculated from 10 day survival data according to [46,47]. The LD_50_ was ~10^3^ CFU per C57BL/6 mouse, and a similar value was used for BALB/c mice.

### 2.5. Fitness of NC983 In Vivo

To determine the ability of NC983 to colonize different tissues, groups of four C57BL/6 female mice (aged 6–8 weeks) were inoculated with ~5 × 10^7^ CFU/mouse of either the parental strain (14028s) or the vaccine strain (NC983). Mice were euthanized at indicated time points and viable *S.* Typhimurium or NC983 within the spleen and liver were determined as described above.

In another experiment, the competitive index (CI; [48]) for NC983 Rif ^R^ (i.e., NC1190) and the virulent 14028s Kan ^R^ (i.e., NC1040) was determined. Fourteen mice, C57BL/6 as above, were given an oral dose of 7.8 × 10^6^ and 7.8 × 10^6^ of NC1190 (NC983-Rif ^R^) and NC1040 (14028s-Kan ^R^), respectively. At four days post infection (dpi), mice were euthanized and the bacterial burden in homogenized tissues was determined by plating each sample on XLT4-MOPS agar plates containing 100 µg/mL rifampicin (to enumerate NC983), and XLT4-MOPS agar plates containing 65 µg/mL kanamycin (to enumerate 14028s).

### 2.6. Vaccination and Challenge Protocols

The mice were subjected to the vaccination protocol shown in Figure 1. The vaccination and boosting doses were determined in preliminary studies. Each vaccinated or challenged mouse received 100 µL of the appropriate cell suspension (see above) by oral gavage. Control mice received an equal volume of the PBS solution.

In vaccination experiment no. 1, C57BL/6 mice were given a vaccination dose (~10^7^ CFU/mouse) and at 14 days post vaccination (dpv) they received a boosting dose (~10^8^ CFU/mouse). Earlier experiments indicated that vaccine and boosting doses of ~10^7^ and ~10^8^, respectively, were optimal for the immune response (data not shown). At 21 days post boosting, (Equals 35 dpv) all mice were challenged with a dose of 100× the LD_50_ (i.e.,10^5^ CFU/mouse) of the virulent *S.* Typhimurium strain (NC1189 (Rif ^R^) and disease symptoms were monitored using the body condition scoring (BCS) as described in [49]. A BCS score of 2 indicates that the animal is under-conditioned and is considered moribund. A BCS score of 2 is observed in mice that exhibit segmentation of vertebral column with detectable pelvic bones. A BCS score of 4–5 indicates a healthy mouse that does not exhibit lack of grooming, eating/drinking, nesting, and other functions of active mice

Mice that survived until 69 dpv were re-challenged with a higher dose, 1000 × LD_50_ (i.e., 10^6^ CFU/mouse). At 90 dpv (i.e., 55 and 21 days post first and second challenges, respectively), all mice were euthanized, blood for analysis of the anti-*Salmonella* IgG response was obtained by cardiac puncture. Cardiac puncture blood was collected at the end of vaccine experiment no. 1 in Sarstedt micro tube 1.1 mL Z-gel (Fisher Scientific, Pittsburgh, PA, USA, catalog no. 50-809-211), and serum was collected according to the supplier instructions. The bacterial burden of the challenge strain (NC1189) was determined within the colon, spleen, and liver following homogenization of tissues and plating on buffered XLT4-MOPS agar plates containing 100 µg/mL rifampicin as described previously [45].

In vaccination experiment no. 2, (BALB/c) mice were given the vaccine, boost, and challenge doses (C1 and C2) as described above and in Figure 1 to measure the antibody response in BALB/c mice to the vaccine strain NC983 in a longitudinal manner, venous blood was obtained through tail bleeding one day prior to: vaccination (before V), boosting (before B), and the first challenge (before C1; Figure 1). 

### 2.7. Measurement of the Anti-Salmonella IgG Response by Elisa

Blood (from venous or cardiac puncture) was allowed to clot at room temperature (~20 min) before centrifuging at 20,000× *g* for 15 min at 4 °C and the supernatant (i.e., serum) was used to measure the anti-*Salmonella* IgG response.

To determine the end-point titers for detection of anti-*Salmonella* antigen, strain 14028s Kan ^R^ (i.e., NC1040) was grown without shaking (still) overnight in LB. Cells were centrifuged, washed with PBS, and concentrated in PBS to an OD_600_ of 5 and sonicated on ice for 10 cycles (i.e., 15 s on and 30 s off) using a 20 KHz Heat Systems-Ultrasonics, Inc sonicator (Plainview, NY, USA), model W-370—set at 50% of its max output. Cell debris were removed by centrifugation at 20,000× *g* for 15 min and the supernatant (cell-free extract, CF-Ext) was used as the *Salmonella* antigen. The protein concentration in the CF-Ext was determined using the Biorad Protein Assay Dye Reagent Concentrate according to manufacturer’s specifications (Biorad; Hercules, CA, USA).

Proteins from the cell-free extracts were diluted in ELISA coating buffer (50 mM carbonate-bicarbonate, pH 9.6; Sigma-Aldrich, St. Louis, MO, USA) to 250 µg/mL. One hundred µL of the solution was added to each well (25 µg) of a Corning 96-well EIA/RIA clear flat bottom polystyrene microplate (product no. 3361) and the plate was incubated overnight at 4 °C. The following day the solution in each well was removed and wells were washed three times with 200 µL wash solution (50 mM Tris base, 0.14 M NaCl, 0.05% Tween 20, pH 8.0). After washing, the wells were blocked for 15 min with the addition of 200 µL of Super Block (ScyTek Laboratories, Inc.; Logan, UT, USA). Serum samples from mice were 2-fold serially diluted in antibody buffer (50 mM Tris, 0.14 M NaCl, 1% BSA) and 100 µL of each dilution was added to wells in duplicate. Plates were incubated at room temperature for 2 h and washed as described above. Secondary antibody (peroxidase-conjugated goat anti mouse IgG; Jackson ImmunoResearch Laboratories; West Grove, PA, USA) was diluted in antibody buffer to 1:10,000 and 100 µL was added to each well. Plates were incubated at room temperature for 2 h and washed as described above. 100 µL of HRP substrate, 1-Step™ Ultra TMB-ELISA Substrate Solution (ThermoFisher Scientific; Waltham, MA, USA), was added to each well and incubated at room temperature for 15 min. The reaction was terminated by the addition of 100 µL of 2 M H_2_SO_4_ and the absorbance at 450 nm was recorded with a multi-mode plate reader (BioTek Synergy HTX; BioTek Instruments, Inc., Winooski, VT, USA). Mean absorbance values were plotted against the Log_2_ of the reciprocal dilution. A multiple *t*-test with a 5% false discovery rate (FDR) post-hoc test with multiple comparisons was used to determine significance. Significance was determined by comparing the mean OD_450_ values of naïve litter mate controls (vaccination experiment no. 1) or against the pre-vaccination values (vaccination experiment no. 2). Figures and statistical analysis were accomplished using GraphPad Prism v7.03.

### 2.8. Measurement of the Anti-Salmonella IgG Response by Immunoblot

Strain 14028s was grown as described above, centrifuged, washed, and the cell pellets were suspended in Laemmli sample buffer. Samples were denatured by boiling. Approximately 2 × 10^8^ cells were loaded per lane and samples were separated by size on 15% acrylamide gels (SDS-PAGE) and transferred to 0.2 μM nitrocellulose membranes (Bio-Rad, Hercules, CA, USA). Immunoblotting was performed as described previously [50,51]. Briefly, membranes were stained with Ponceau S (0.1% Ponceau S (*w*/*v*), 1% acetic acid) to ensure equivalent loading of samples. For immunoblotting, membranes were blocked in a blocking buffer (PBS containing 0.05% Tween-20 and 1% powered non-fat milk, pH 7.4) and probed with serum from BALB/c mice (primary antibody at 1:1000 for 3 h). Membranes were washed three times with the blocking buffer and probed with secondary antibody (peroxidase-conjugated goat anti mouse IgG; Jackson ImmunoResearch Laboratories; West Grove, PA) at 1:5000 for 3 h. Membranes were washed three times with Tris-NaCl (50 mM Tris, 200 mM NaCl, pH 7.6) and detection of horseradish peroxidase activity was determined in Tris-NaCl using 4-chloro-1-napthol (4CN; dissolved in methanol) and H_2_O_2_ (Thermo Fisher Scientific; Waltham, MA, USA).

### 2.9. Statistical Analysis

For survival plots, Log-ranked (Mantel-Cox) test was applied using Graph Pad Prism v. 7.03. For statistical analysis of anti-*Salmonella* IgG, a multiple *t*-test with a 5% false discovery rate (FDR) post-hoc test with multiple comparisons (Graph Pad Prism v. 7.03) was used to determine significance. In all cases *p*-values < 0.05 were considered significant.

### 2.10. Ethics Statement

This study was carried out in strict accordance with the recommendations in the guide for the Care and Use of Laboratory Animals of the National Institutes of Health. All mice were maintained and euthanized according to a protocol (no. 15-035-B) approved by the Institutional Animal Care and Use Committee (OLAW no. D16-00214).

## 3. Results

### 3.1. Strain NC983 Exhibits a Fitness Defect in the Colonization of the Liver and Spleen

We determined the kinetics of liver and spleen colonization for strain NC983 and the challenge virulent strain 14028s—Rif ^R^ (NC1189). At 1 dpv, 3 out of the 4 mice had detectable levels of NC983 in the spleen and liver tissues (Figure 2A). At 2 dpv, all mice had quantifiable levels of NC983, but at 4, 8 and 15 dpv there was at least one mouse at each time point with undetectable levels of NC983 (i.e., below the detection level) (Figure 2A). At 35 dpv, one mouse had detectable NC983 in the splenic tissues (Figure 2A). On the other hand, the kinetics of liver and spleen colonization by the virulent challenge strain, 14028s-Rif ^R^ (NC1189), showed a different pattern (Figure 2B). At days 1, 2, 4 and 6 post infection, mice were euthanized, and the bacterial burden was determined. By 4 dpi, all mice had concentrations of the challenge strain that were >10^4^ CFU/g tissue; and at 6 dpi, concentrations of 14028s-Rif ^R^ (NC1189) reached ~10^7^ CFU/g in all mice (Figure 2B). No further time-point data were collected because the mice had a body condition score (BCS) ~2 and were euthanized.

Clearly, at days 1 and 2 post inoculation the kinetics of colonization of the liver and spleen by the vaccine strain was like that of the wild-type parent strain. However, at 4 dpv and beyond, the vaccine strain showed much weaker colonization of the liver and spleen than the wild type.

This finding was confirmed by the data from the competitive index study (Figure 2C). In this type of assay, the fitness of the vaccine strain in the different murine tissues is simultaneously compared to that of the wild-type strain in the same animal [48] and Figure 2C. The Log_10_ of the competitive index (CI) for the vaccine strain (NC1190) versus the challenge strain NC1040 was variable between the different animals in the colon tissue sites, with an average fitness defect of ~1.4 orders of magnitude (Figure 2C). However, within the liver and spleen for all mice there was a clear fitness defect of ~3.7 orders of magnitude (i.e., 5000-fold reduction) between the vaccine strain and the wild-type virulent strain (NC1040) (Figure 2C). The kinetics and CI data (Figure 2) indicated that strain NC983 has a general fitness defect in the mice but showed a clear and profound defect within the liver and spleen.

### 3.2. Strain NC983 Is a Live Attenuated Salmonella Strain that Protects against Virulent S. typhimurium and Is Immunogenic in Mice

Previous work demonstrated that strain NC983 was unable to cause lethal infection in C57BL/6 mice when inoculated through either peroral or intraperitoneal routes [44]. This evidence suggested that NC983 may be attenuated in mice and warranted further studies to test its ability to confer protective immunity in mice. Therefore, a vaccination protocol was developed to test the ability of NC983 to protect against challenge with virulent *S.* Typhimurium (Figure 1). This protocol utilized oral inoculation (i.e., vaccination and boost) of mice with either strain NC983 (vaccine group) or a PBS control. At 35 days post-vaccination (dpv), all mice were challenged with the virulent strain of *S.* Typhimurium ATCC 14028s, as outlined in Materials and Methods and Figure 1. The percent survival of mice was recorded for the duration of the study, bacterial burden of the challenge strain in vaccinated mice and anti-*Salmonella* IgG in vaccinated mice were determined at the end of the study (Figure 3).

The data showed that by day 17 post challenge, all control mice (*n* = 3) had died or required euthanasia (Figure 3A). Although one mouse of the vaccinated group was found dead after the boosting, the remaining mice (*n* = 5) exhibited 100% survival post the two challenges (Figure 3A). At 90 dpv, all mice were euthanized, and samples were processed to determine the bacterial burden of the WT virulent *S.* Typhimurium strain in the vaccinated mice (Figure 3B). The level of the vaccine strain (NC983) was undetectable in all tissues and the colon of these mice (data not shown). However, the challenge strain was found in quantifiable levels in colon samples from two mice. In addition, three splenic samples and two liver samples exhibited levels of the challenge strain between 10^2^ and 10^3^ CFU/g (Figure 3B). When the anti-*S.* Typhimurium IgG levels were measured from these mice at 90-dpv, the mean endpoint titer was 1: 256,000 (FDR adjusted *p* value = 0.044; Figure 3C).

The vaccination protocol was repeated (vaccine experiment no. 2) with another *Salmonella* sensitive strain of mice (BALB/c) to ensure results from C57BL/6 were not strain specific. In this experiment, a preliminary LD_50_ for the BALB/c showed that it was slightly lower than that of the C57BL/6 mice. However, to be on the safe side, we used the same LD_50_ as that for the C57BL/6. Also, the responses of individual mice were measured over time in a longitudinal approach (Figure 4). At 35 dpv, the BALB/c mice received on oral dose of the virulent strain, NC1040 (6.4 × 10^4^ CFU/mouse). By 7 days post challenge (C1, or 42 dpv), all control mice had died or required euthanasia (Figure 4A).

At 21 days post challenge (56 dpv), one vaccine group mouse had to be euthanized; however, the remaining five mice survived another challenge dose (C2; 1.2 × 10^6^ CFU/mouse) (Figure 4A). At the end of the experiment (90 dpv), the bacterial burden of the challenge strain NC1040 was determined. Three mice (mouse 1, 2 and 5) had detectable levels of the challenge strain in all three examined tissues (colon, spleen, and liver). However, we did not detect the challenge strain in the colons of mouse 3 or 4. In addition, mouse 3 contained the challenge strain in the spleen and liver tissues, but mouse 4 had no challenge strain in any examined sites (Figure 4B).

To measure the IgG response to NC983, serum was obtained through tail bleeding at one day prior to vaccination, boosting, and the first challenge (Figure 1). The mean endpoint titer taken before the boosting dose (13 dpv, before B) was 1:100 (dotted line, FDR adjusted *p* value = 0.031; Figure 4C). The mean endpoint titer after vaccination and boosting was 1: 3200 (solid line, FDR adjusted *p*-value = 0.018; Figure 4C).

Serum samples from individual mice were probed against whole cell lysates from *S.* Typhimurium by immunoblotting (Figure 4D). The data showed that serum taken before vaccination had no cross reactivity to *S.* Typhimurium antigens (lanes marked V). However, after vaccination 3 out of 5 mice showed a cross reactivity band at ~40 kDa (lanes marked B); but after the second inoculation (boosting) all mice showed multiple cross reactivity bands (lanes marked C1). Clearly, there was significant increase in cross reactivity to *S.* Typhimurium antigens from all mice at the C1 time point (i.e., just before the challenge) (Figure 4D). Indeed, further studies are needed to identify the different *S.* Typhimurium antigens reacting with the antibodies produced in the immunized mice.

## 4. Discussion

*S. typhimurium* is a significant global health problem. In the USA, it is leading agent that causes bacterial foodborne illness [1]. In addition, highly invasive Typhimurium and NTS isolates are becoming increasingly problematic in specific areas of the World [52,53,54,55]. Therefore, there is a demand for effective preventative measures. Here we demonstrated the effectiveness of the live attenuated *S.* Typhimurium strain (NC983) that fully protected two genetically distinct mice backgrounds from challenge with virulent *S. typhimurium* (Figure 3A and Figure 4A). Vaccinated mice exhibited an anti-*Salmonella* IgG response (Figure 3C and Figure 4C) and strain NC983 was sporadically capable of reaching systemic tissues sites (Figure 3B and Figure 4B) while exhibiting a pronounced fitness defect in the liver and spleen (Figure 2). Collectively, these results support the conclusion that strain NC983 is an immunogenic, live attenuated vaccine strain.

Strain NC983 is derived from the highly virulent strain 14028s (American Type Culture Collection strain, ATCC 14028s; a smooth-colony variant derived from CDC60-6516) that was isolated in 1960 from samples of hearts and livers of 4-week-old chickens [56,57].

In our initial studies with strain NC983 [44], it was determined that it was attenuated either through oral or intraperitoneal routes of infection. It was suggested that this attenuation was due a mutation in the anaerobic regulator, *fnr* [44]; however, subsequent work has determined that strain NC983 contains a large deletion of the genome. Sequencing the genome of this strain revealed a deletion that removed base pairs 1,737,878 to 1,764,448 from the genome of 14028s [58]. This deletion extends from the 3′ end of *STM14_1981* to the 5′ end of *STM14_2007* (*fnr*), which has effectively inactivated 26 known protein coding genes. The missing segment has been replaced with a remnant from the transposable element from Tn10. At present, single-gene knockouts of candidate genes from this region have been tested in vivo (*STM14_1997* (*ynaF*), *STM14_2002* (*zntB*), *STM14_2007* (*fnr*)), but these mutants have been unable to duplicate the phenotype of NC983. Additional studies are required to determine the precise contribution of gene(s) within this region toward infection. However, because of this large deletion, it is less likely that this strain will undergo a reversion to virulence within the host. Because this genetic region is conserved within the *S. enterica* genomes sequenced to date, our results suggest that the genetic mutation within NC983 may provide a guide for construction of other live attenuated serovars and strains. In addition, strain NC983 may be an effective platform for the delivery of heterologous antigens.

Macrophages appear to play an important role in the host–pathogen interactions of mouse and *S.* Typhimurium [59,60,61,62]. NC983 is sensitive to the antimicrobial response of macrophages that contain a functional NADPH oxidase enzyme complex (Nox). Although NC983 exhibited a survival defect within Nox competent macrophages, it was capable of surviving within macrophages isolated from gp91*^phox-/-^* mice [44]. Moreover, strain NC983 caused lethal infection in gp91*^phox-/-^* mice following infection through the intraperitoneal route [44]. Thus, NC983 appears metabolically fit within the murine host. Even though the contribution of the Nox enzyme complex to combating *S.* Typhimurium infection has focused on the innate cell production of reactive oxygen species (ROS) to kill the pathogen, recent work has indicated that Nox-dependent production of ROS by B cells may be important. B cells are capable of phagocytosing *S.* Typhimurium with subsequent activation of CD4^+^ T cells that are critical in the development of an anti-*Salmonella* response [63,64]. In addition, the Nox complex is necessary for efficient antigen presentation by human B cells in a MHC class II context to CD4^+^ T cells [65]. Finally, the role of B cells in the protective response against *S.* Typhimurium has been shown to be independent of antibody production, but dependent on the promotion of the T cell response to infection [66]. Because of the production of anti-*Salmonella* IgG (Figure 3C and Figure 4C) it suggests that our vaccination protocol with NC983 in mice is adequate for the B cell mediated activation of the T cell response. However, additional experiments are required to definitively ascertain the role of NC983 in modulating the naïve T cell population in vivo.

Live attenuated *Salmonella* strains have been tested in humans and other animals. An initial Phase I trial aimed to use a live, attenuated strain to treat metastatic melanoma [67]. Following infusion, studies indicated the strain was tolerated at doses <3 × 10^8^ CFU/m^2^. Earlier work indicated that a live attenuated strain from serovar Typi (causative agent of Typhoid fever) was tolerated in adult volunteers via oral route of administration [68]. Notably, oral administration of another live attenuated Typhi strain has completed Phase II studies with demonstrated safety and immunogenicity [69]. Since genetic mutations that promote safety in serovar Typhi may not be directly translatable to NTS [70], studies may require serovar-specific approaches to determine the genetic conditions that promote live attenuation. Animal studies in dogs, poultry, and swine have demonstrated the safety and efficacy of live attenuated *Salmonella* strains [71,72,73]. Therefore, the study of live attenuated strains in vaccine development may provide solutions to public health concerns.

This work demonstrated that strain NC983 was immunogenic and protective in two mouse backgrounds from virulent Typhimurium challenge. The genetic mutation in NC983 could be recapitulated in different serovar genomes to produce serovar-specific live attenuated vaccines. Work is in progress to identify the critical genetic mutation within this region that confers the vaccine phenotype and introduce this mutation in different serovars of *S. enterica*. Because this genetic region is conserved within the *S. enterica* genomes sequenced to date our results suggest that the genetic mutation within NC983 may provide a guide for construction of live attenuated strains. In addition, strain NC983 may be an effective platform for the delivery of heterologous antigens that require a CD4^+^ T cell response for protection.

## 5. Patents

U.S. Provisional Patent—Engineered *Salmonella* Serovar Typhimurium Strains, Compositions Thereof, and Methods of Use—Serial No. 62/368,507-Issued 29 July 2016.

Attenuated FNR Deficient Enterobacteria

U.S. Patent No: 8435,506—Issued 7 May 2013

Attenuated FNR Deficient Enterobacteria

U.S. Patent No: 8101,168—Issued 24 January 2012

## Figures and Tables

**Figure 1 vaccines-08-00646-f001:**
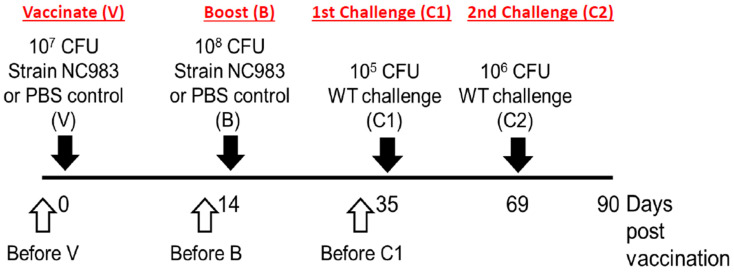
Schematic representation of the vaccination protocol. Female mice 6–8 weeks of age (C57BL/6 or BALBc) were used in these studies. The following symbols represent the different treatments at the specified time points: V = vaccination; B = boosting; C1 = first challenge; and C2 = second challenge. In each study, two groups of mice (vaccinated and control) were subjected to the outlined protocol. Each mouse in the vaccinated group received 100 µL containing ~10^7^ CFU of NC983, while each mouse in the Naïve control group received an equal volume of a PBS solution. In vaccination experiment no. 2, venous blood from the tail was collected one day prior to V, B, and C1—as indicated by the open arrows for assaying anti-*Salmonella* IgG response.

**Figure 2 vaccines-08-00646-f002:**
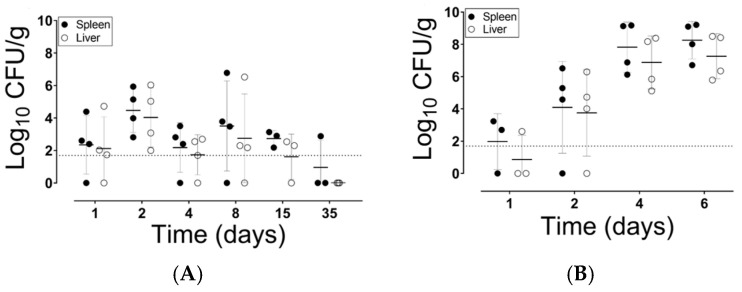
Kinetics of the systemic colonization and competitive fitness of strain NC983 and virulent *S.* Typhimurium. (**A**) Six groups of 4 female C57BL/6 mice (a total of 24 mice) were inoculated with 5 × 10^7^ CFU of NC983. At 1, 2, 4, 8, 15 and 35 days post vaccination, 4 mice were euthanized and the bacterial burden of NC983 in the spleen (filled circles) and liver (open circles) was determined. Each point is an individual mouse and the mean ± 1 standard deviation is shown. For both (**A**,**B**), the dash line shows the limit of detection. (**B**) Four groups of 4 C57BL/6 mice (a total of 16 mice) were inoculated with 5 × 10^7^ CFU of NC1189 (virulent Typhimurium). At 1, 2, 4 and 6 days, 4 mice were euthanized and the bacterial burden in the spleen (filled circles) and liver (open circles) tissue was determined as in (**A**). (**C**) Strain NC983 exhibits a fitness defect within the spleen. Four 6–8-week-old female C57BL/6 mice were orally inoculated with a mixture containing 9.1 × 10^6^ of NC1190 (NC983-Rif ^R^) and 8 × 10^6^ of NC1040 (ATCC 14028s-Kan ^R^). At four days post infection (dpi), mice were euthanized and the bacterial burden in homogenized tissues was determined, as described in Materials and Methods. The competitive index (CI; [48]) was calculated using the following equation: (NC1190_OUT_/NC1040_OUT_)/(NC1190_IN_/NC1040_IN_). Each data point is the log_10_ of the CI from a single mouse and tissue site.

**Figure 3 vaccines-08-00646-f003:**
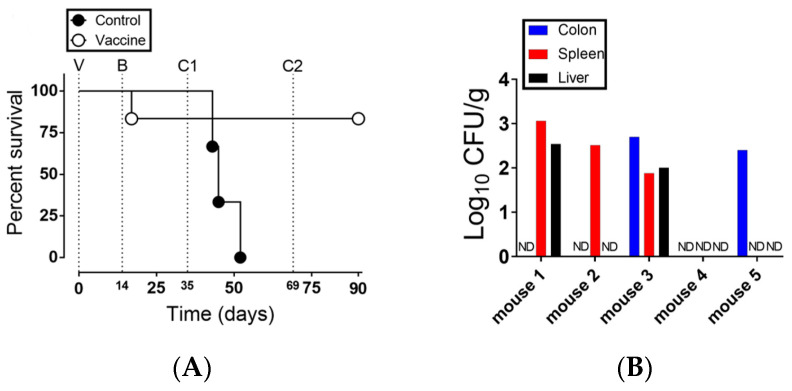
Vaccination of C57BL/6 mice with strain NC983 protects against challenge with virulent *S*. Typhimurium. (**A**) Survival of vaccinated C57BL/6 mice (experiment no. 1). Two groups of mice (control group, *n* = 3; and vaccine group, *n* = 6) were subjected to the vaccination protocol outlined in Figure 1. Mice in the vaccine group received oral doses of NC983 equivalent to 6 × 10^7^ CFU/mouse and 5 × 10^8^ CFU/mouse for vaccination and boosting, respectively. Mice in the control group received PBS. Mice in both groups were challenged with *S.* Typhimurium strain (NC1189) at 2 × 10^5^ CFU/mouse (200 × LD_50_); and surviving mice were challenged again with a higher dose of 4 × 10^6^ CFU/mouse (4000 × LD_50_). Survival of mice was monitored over time and expressed as percent. Statistical comparison of survval curves using Log-ranked (Mantel–Cox) test showed a *p*-value of 0.0397. (**B**) Bacterial burden of the challenge strain (NC1189) in the vaccinated mice at termination of the experiment in (**A**). Tissue samples were homogenized and plated on XLT4 agar plates containing 100 µg/mL of rifampicin and incubated at 37 °C for 24 h to enumerate bacteria. Rif ^R^ H_2_S producing colonies were counted and are expressed as log_10_ of the CFU/g of tissue sample. (**C**) Production of anti-*S.* Typhimurium IgG in vaccinated mice. At the end of experiment no. 1 serum was collected from the surviving animals and assayed for anti-*Salmonella* IgG, as described in Material and Methods. The last reciprocal dilution with mean OD_450_ values that were significantly different than the negative control serum was considered the endpoint dilution. The solid line shows the mean endpoint dilution (1: 256,000; FDR adjusted *p*-value = 0.044). A multiple *t*-test with a 5% false discovery rate (FDR) post-hoc test with multiple comparisons was used to determine significance. Significance was determined by comparing the mean OD_450_ values against naïve litter mate controls (shown as a dotted line).

**Figure 4 vaccines-08-00646-f004:**
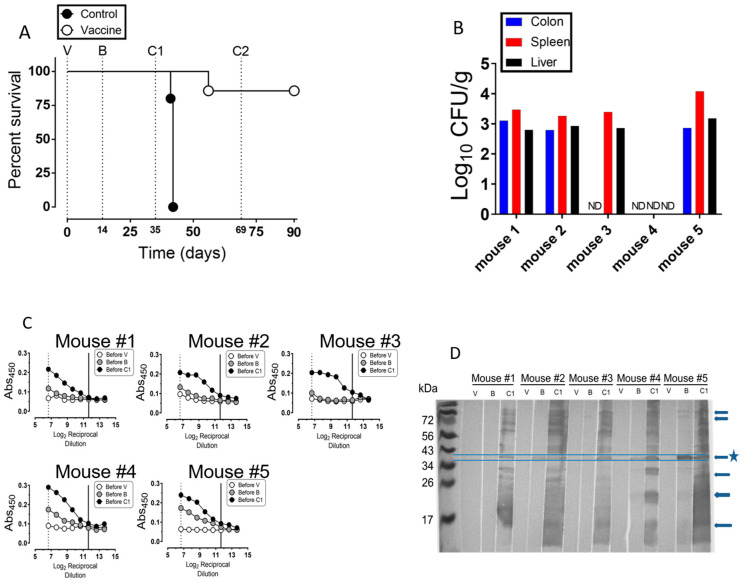
Strain NC983 induces an IgG response and is protective in BALB/c mice against virulent *S.* Typhimurium challenge. (**A**) Survival curve of vaccinated BALB/c mice (experiment no. 2). Two groups of mice (control group, *n* = 5; and the vaccine group, *n* = 6) were subjected to the vaccine protocol described in Figure 1 and Figure 3A. At V and B, each mouse in the vaccinated group received 9 × 10^6^ CFU/mouse and 1 × 10^8^ CFU/mouse of NC983, respectively, while the control mice received PBS. At C1 and C2, mice were challenged with the virulent strain NC1040 at 6 × 10^4^ and 1.2 × 10^6^, respectively. Survival of mice was monitored over time and is expressed as percent. Statistical comparison of survival curves using Log-ranked (Mantel–Cox) test showed a *p*-value of 0.0009. (**B**) Bacterial burden of the virulent challenge strain (NC1040) in vaccinated mice at termination of experiment no. 2. Tissue samples were homogenized and plated on XLT4 agar plates containing 65 µg/mL of kanamycin and incubated at 37 °C for 24 h to enumerate bacteria. Kan ^R^ H_2_S colonies were counted and are expressed as log_10_ of the CFU/g of tissue sample. (**C**). Production of anti-*Salmonella* in response to vaccination and boosting. One day prior to V, B, and C1 (open arrows, in Figure 1) venous blood from the tail was removed to obtain the basal, vaccine-induced, and boosting-induced anti-*Salmonella* IgG response, respectively. Twenty-five micrograms of NC1040 protein were added to each well, serum samples prepared from tail’s venous blood one-day before vaccination, boosting and challenge were two-fold serially diluted, and analyzed in duplicate. Data shown are the log_2_ of the reciprocal dilution. Statistical significance was determined by comparing the mean OD_450_ values against the before V values at each dilution. The last reciprocal dilution with mean OD_450_ values that were significantly different than the before V values were considered the endpoint dilution. The dotted line (1:100; FDR adjusted *p* value = 0.031) and solid line (1: 3200; FDR adjusted *p* value = 0.018) show the endpoint dilutions for the before V and before B doses, respectively. A multiple *t*-test with a 5% false discovery rate (FDR) post-hoc test with multiple comparisons was used to determine significance. Significance was determined by comparing the mean OD_450_ values against the pre-vaccination values (vaccination experiment no. 2). (**D**) Detection of the anti-*S.* Typhimurium IgG response by immunoblotting. The equivalent of 2 × 10^8^ cells of whole-cell lysate from strain 14028s was loaded per lane and samples were separated by size on 15% acrylamide gels. Following transfer, membranes were blocked and probed with serum from individual BALB/c mice. Serum samples were obtained by tail bleeding at V (1 dpv), B (13 dpv), and C1 (34 dpv) to determine the host IgG response. Membranes were probed with secondary antibody (anti-mouse IgG conjugated to HRP) and detection of horseradish peroxidase activity was determined with 4-chloro-1-napthol and H_2_O_2_, as described in Materials and Methods. Arrows ←, represent antigen-antibody complexes; Stared arrow ←*, represents early antigen-antibody complex.

**Table 1 vaccines-08-00646-t001:** Bacterial strains used in this study

Strain	Genotype ^a^	Source
*Salmonella enterica* serovar Typhimurium 14028s	Wild-Type	ATCC ^b^
NC983	Fusaric Acid Resistant	[44]
NC1040	ATCC 14028s *fnr*’:*ha* (Kan ^R^)	[45]
NC1189	ATCC 14028s (Rif ^R^)	This study
NC1190	NC983 (Rif ^R^)	This study

^a^ Rif ^R^ (rifampicin resistant) and Kan ^R^ (kanamycin resistant). ^b^ ATCC (American Type Culture Collection).

## Data Availability

The datasets generated and analyzed during the current study are available from the corresponding author on reasonable request.

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
