# Peer review of "Attenuated Salmonella enterica Serovar Typhimurium, Strain NC983, Is Immunogenic, and Protective against Virulent Typhimurium Challenges in Mice"

_vaccines, 2020, doi:10.3390/vaccines8040646_

Round 1

Reviewer 1 Report

Article must be carefully checked prior of being really reviewed. Some comments below:

Typos and please review use of achronims (i.e. lines 33 you use USA, then in line 34 US, etc).

In the introduction you do not mention at all candidate vaccines against NTS that are not live attenuated bacteria(i.e. glicoconjugates, GMMA-based vaccines, etc). Please mention/discuss them.

ELISA methodology: coating material used looks not characterised accordingly to the protocol. Showing a simple SDS-PAGE for protein patter and a Silver Staining to determine LPS presence might be helpfull. This is to understand if the cell free suspension is containing the majority of surface antigens of Salmonella or not (and similarly if oterwise citoplasmic proteins . If ELISA is not performed with well characterised antigens the use of heat killed bacteria might be a good alternative.

Between page 6 and 7 seems that part of the text is missing (see line 271). It is therefore impossible to provide a full revision of the manuscript.

Discussion is completely missing any potential tolerability of the vaccine in humans and animals.

Author Response

Comments and Suggestions for Authors

  • Article must be carefully checked prior of being really reviewed. Some comments below:

The manuscript was carefully checked by all authors. Below are our responses to the Reviewer Comments:

  • Typos and please review use of achronims (i.e. lines 33 you use USA, then in line 34 US, etc).

We thank the reviewer to pointing out this discrepancy. This has been corrected in  the revised manuscript.

  • In the introduction you do not mention at all candidate vaccines against NTS that are not live attenuated bacteria(i.e. glicoconjugates, GMMA-based vaccines, etc). Please mention/discuss them.

We thank the reviewer for pointing out this important deficit in the Introduction. We had not considered the impact of alternative approaches to NTS vaccine development. Although the focus of this work is on the utility of the live attenuated strain, we have added a paragraph in the Introduction that cites relevant work regarding alternative approaches to NTC vaccine development. Please refer to the revised manuscript.

  • ELISA methodology: coating material used looks not characterised accordingly to the protocol. Showing a simple SDS-PAGE for protein patter and a Silver Staining to determine LPS presence might be helpfull. This is to understand if the cell free suspension is containing the majority of surface antigens of Salmonella or not (and similarly if oterwise citoplasmic proteins . If ELISA is not performed with well characterised antigens the use of heat killed bacteria might be a good alternative.

Please note that the source of Salmonella antigens for the ELISA and Western were from bacterial cultures grown with the same conditions. Although the coating reagent was cleared of cell debris after sonication, the IgG response from individual mice demonstrate a similar measurement regardless of the source of Salmonella antigens (i.e., the ELISA data and Western indicate an increase in the IgG response for both antigen preparations). Importantly, the IgG response was measured from mice that survived challenge with the virulent 14028s strain. The data indicated there was a clear increase in the IgG antibody pool against the Salmonella antigens used. In addition, this response was associated with survivability even when mice were challenged at 100X to 1000X the LD50 for both mice strains.

Although the characterization of antigen(s) that provide protection are of interest, this work set to establish the live attenuated strain as a candidate for future studies. To accomplish this, the work focused on viability of mice after challenge with moderate to high doses of virulent Salmonella.

  • Between page 6 and 7 seems that part of the text is missing (see line 271). It is therefore impossible to provide a full revision of the manuscript.

We checked the formated manuscript and found nothing is missing! Furthermore, the statement about (see line 271) is not clear. Line 271 is within the figure legend of Figure 3!

  • Discussion is completely missing any potential tolerability of the vaccine in humans and animals.

We thank the reviewer for pointing out this omission. Please refer to the revised manuscript, where we add a paragraph about the potential tolerability of attenuated vaccines in animals and humans. We hope that the reviewer finds this added paragraph sufficient to address the concerns.

Reviewer 2 Report

The authors show preliminary support for protection against wild-type Salmonella after vaccination with an attenuated strain. Although they show that survival and bacterial burden are imporved upon vaccination there are several points to be addressed:

1) Were the mice pretreated with antibiotics?

2) Is protection lost in B or T cell deficient mice?

3) Does vaccination also protect from high dose infections?

4) How many days after vaccination can be protection achieved?

5) What are potential mechanisms for protection? Does a T or B cell transfer from vaccinated mice protect, for example, naive mice from infection?

6) Is protection dependent on persistence of the attenuated strain in the gut?

7) Do heat killed bacteria exhibit a similar effect?

Author Response

The authors show preliminary support for protection against wild-type Salmonella after vaccination with an attenuated strain. Although they show that survival and bacterial burden are imporved upon vaccination there are several points to be addressed:

  • Were the mice pretreated with antibiotics?

Mice were not pretreated with antibiotics prior to Vacination nor Infection. In this study, we were not interested in studying the rffects of perturbation of the intestinal microbiota on the vaccination or the infection. 

  • Is protection lost in B or T cell deficient mice?

We thank the reviewer for acknowledging the importance of understanding the contribution of the adaptive immune response to understanding protection provided by the vaccine strain. Additional studies are needed to address this question. The contribution of B cells and or T cells can be complex. Multiple murine lines are required to fully understand how these cells contribute to the protection in response to the vaccine strain.

  • Does vaccination also protect from high dose infections?

Yes, the vaccination protocol was design to test 100X and 1,00X the LD50 value. However, in figure #3 we presented data showong that the vaccine provided protection against 4,000X the LD50. Thus, the protection that was provided by the vaccine strain, following vaccination and boosting, demonstrates efficacy at high dose challenges with the virulent strain.

  • How many days after vaccination can be protection achieved?

We thank the reviewer for asking this question. This experiment has not been conducted yet. This would be a beneficial data set since the results would determine if the vaccine strain can provide long term immunity against subsequent challenge.

  • What are potential mechanisms for protection? Does a T or B cell transfer from vaccinated mice protect, for example, naive mice from infection?

We thank the reviewer for posing this question. This was not done in this study. It is on our to-do list for future studies. We believe that this is a good experiment to conduct. Our working hypothesis is that the vaccine strain protects against challenge via the inability to persist within innate cells while stimulating a CD4+ T cell response. Isolation of specific T and B cells populations from immunized mice prior to challenge would contain pools of candidate cells for testing.

  • Is protection dependent on persistence of the attenuated strain in the gut?

No, protection is not dependent on persistance of the attenuated strain, NC983, in the gut. Line 291 of the original manuscript stated " The level of the vaccine strain was undetectable in all tissues and the colon of these mice (data not shown).

In general, the vaccine strain exhibits a profound defect in the survivability within systemic tissue sites (liver/spleen). At day 4 post administration of NC983, Colonization of the intestines was reduced ~.one log compared to the virulent strain Fig. 2C.  At day 35 (before the first challenge) the vaccine strain was undetectable within the gut (as stated above). 

  • Do heat killed bacteria exhibit a similar effect?

This experiment has not been performed. The cultivation conditions of the bacteria prior to heat killing may have a direct impact on the efficacy of this approach. However, previous published report (Ref. #6) demonstrated that heat killed Salmonella are not as protective as "live attenuated bacteria".

Round 2

Reviewer 1 Report

None

Author Response

Comments and suggestion to authors: None

Thanks

Reviewer 2 Report

The concerns should be at least partially addressed by the suggested experiments. Thus, the experiment with the heat killed bacteria should be performed as well as the question for how many days after application of the vaccinia strain protection can be observed.

Author Response

Reply to Reviewer's Comments:

Regarding “the experiment with the heat killed bacteria should be performed”

We appreciate the reviewer’s question about the use of heat-killed bacteria as vaccine. We did not carry this type of study for the following reasons:

1) The scope of the study is attenuated Salmonella as a vaccine/ a vector.

2) Previous studies [e.g., Collins, FM and Carter, PB; 1972, Infection and Immunity 6: 451-458] – demonstrated that “oral vaccination of CD-1 mice with heat-killed suspension of Salmonella was never able to prevent the development of clinical disease in the challenged mice”.  Also, Hashizume-Takizawa, T; 2015, Int. J. Oral Med Sci 14 (2) (3): 54-60 – concluded that “heat-killed r.Salmonella – Tox C may not be appropriate for use as an oral vaccine or vaccine vector”.

Therefore, according to the Ethical use of animals in research, we could not sacrifice the animals knowing that there is ample evidence in the literature showing that the study is counterproductive.

Regarding “the question for how many days after the application of the vaccine strain protection can be observed”

The main scope of the current study is to demonstrate that this genetic mutant of Salmonella can confer immunity. Indeed, the type of the recommended study is on our to-do-list and could take from 6-12 month to complete. Unfortunately, due to COVID-19 and the fact that we are working remotely, our animal studies are on-hold.  
